# Investigating the Failure Mechanism of p-GaN Gate HEMTs under High Power Stress with a Transparent ITO Gate

**DOI:** 10.3390/mi14050940

**Published:** 2023-04-26

**Authors:** Zhanfei Han, Xiangdong Li, Hongyue Wang, Jiahui Yuan, Junbo Wang, Meng Wang, Weitao Yang, Shuzhen You, Jingjing Chang, Jincheng Zhang, Yue Hao

**Affiliations:** 1Guangzhou Wide Bandgap Semiconductor Innovation Center, Guangzhou Institute of Technology, Xidian University, Guangzhou 510555, China; zfhan@stu.xidian.edu.cn (Z.H.); yuanjh1126@163.com (J.Y.); wjb874085975@163.com (J.W.); mengm10612183@163.com (M.W.); yangweitao@xidian.edu.cn (W.Y.); youshuzhen@xidian.edu.cn (S.Y.); jjingchang@xidian.edu.cn (J.C.); yhao@xidian.edu.cn (Y.H.); 2China Electronic Product Reliability and Environmental Testing Research Institute, Guangzhou 511370, China; 3Key Laboratory of Wide Bandgap Semiconductor Materials and Devices, School of Microelectronics, Xidian University, Xi’an 710071, China

**Keywords:** p-GaN gate HEMTs, transparent indium-tin-oxide, reliability, high power stress

## Abstract

The channel temperature distribution and breakdown points are difficult to monitor for the traditional p-GaN gate HEMTs under high power stress, because the metal gate blocks the light. To solve this problem, we processed p-GaN gate HEMTs with transparent indium tin oxide (ITO) as the gate terminal and successfully captured the information mentioned above, utilizing ultraviolet reflectivity thermal imaging equipment. The fabricated ITO-gated HEMTs exhibited a saturation drain current of 276 mA/mm and an on-resistance of 16.6 Ω·mm. During the test, the heat was found to concentrate in the vicinity of the gate field in the access area, under the stress of V_GS_ = 6 V and V_DS_ = 10/20/30 V. After 691 s high power stress, the device failed, and a hot spot appeared on the p-GaN. After failure, luminescence was observed on the sidewall of the p-GaN while positively biasing the gate, revealing the side wall is the weakest spot under high power stress. The findings of this study provide a powerful tool for reliability analysis and also point to a way for improving the reliability of the p-GaN gate HEMTs in the future.

## 1. Introduction

The field of power electronics has shown considerable interest in GaN material, owing to its excellent properties including high mobility, high breakdown voltage, and wide bandgap [1,2,3,4,5,6,7,8,9,10,11]. For the sake of the reliability of the power system, normally-off devices are preferred in the application. E-mode devices can be realized by several distinct methods including a p-GaN gate [12,13,14,15], recessed structures [16], and fluorine ion implantation [17], where the p-GaN gate is a promising approach that has been adopted by the industry in the fast charging field [18]. The traditional p-GaN gate stack can be modeled as two back-to-back diodes. The Schottky metal/p-GaN junction is reversely biased under a positive gate bias, which often induces to gate degradation and even irreversible failures [19]. To tackle these problems, a PNJ-HEMT was proposed by Hua et al. to improve the gate voltage swing range [20]. Liu et al. proposed an AlGaN/p-GaN/AlGaN/GaN structure to block the carrier injection behavior [21]. GaN HEMTs working under high power stress often suffer fatal failures. However, for the traditional thick gate metal, it is difficult to precisely monitor the breakdown points and the temperature distribution when the devices are stressed by high power, which hinders the research of the device’s reliability [22,23,24]. We note that Wu et al. suggested using indium tin oxide (ITO) instead of the traditional Schottky metal to improve the breakdown voltage of the gate [25,26,27].

Herein, we propose a transparent gate structure using ITO material, which has been maturely applied in the industry field of GaN LEDs, as the gate of HEMTs. The ITO-gated HEMTs were first successfully processed. Then, the temperature distribution in the channel of the device under high power stress was recorded by ultraviolet reflectance thermography equipment. Further, assisted by the luminescence test, the locations of breakdown points were unambiguously determined.

## 2. Materials and Methods

A p-GaN/AlGaN/GaN heterostructure was grown on a 6-inch Si <111> substrate via metal-organic chemical vapor deposition (MOCVD). The structure comprises a 4-µm thick AlGaN buffer layer, an unintentionally-doped 200-nm GaN channel layer, a 0.7-nm AlN layer, a 15-nm Al_0.2_Ga_0.8_N barrier layer, and a 70-nm p-GaN layer doped with a Mg doping concentration of 3 × 10^19^ cm^−3^. The electron mobility extracted using Hall measurement at room temperature was 1495 cm^2^/V·s.

The cross-sectional diagram of the ITO-gated p-GaN gate HEMTs is shown in Figure 1a, where an FIB-SEM photograph of the gate stack as well as the gate field plate is shown in Figure 1b. The device process flow is shown in Figure 2. This process highlights p-GaN selective etching by using Cl_2_/Ar/O_2_-based inductively coupled plasma (ICP), with the AlGaN barrier layer employed as a self-stopping layer. The surface roughness of the etched area was measured to be 0.35 nm via atomic force microscope (AFM) scanning. The device’s source and drain electrodes, Ti/Al/Ni/Au (22/140/50/40 nm), were deposited via electron beam evaporation, and ohmic contacts were achieved by rapid annealing at 865 °C for 30 s in the ambient of N_2_. The device’s passivation layer is a 200 nm SiO_2_ deposited by plasma-enhanced chemical vapor deposition (PECVD). The active region of the device was defined through multiple conditions of N ion implantation isolation to achieve an implantation depth of 300 nm. Finally, the gate region opening was patterned by lithography and then SiO_2_ etching. Afterwards, the ITO was deposited via evaporation as the gate material and patterned by IBE etching. During the ITO deposition, the substrate temperature was maintained at 350 °C. The fabricated devices have a gate length L_G_ of 3 µm, a gate width W_G_ of 100 µm, a gate-to-source distance L_GS_ of 1.5 µm, and a gate-to-drain distance L_GD_ of 16 µm.

## 3. Results

Figure 3a displays the typical I_D_-V_D_ characteristics of the ITO-gated device. At V_DS_ = 10 V and V_GS_ = 6 V, the device exhibits a saturation drain current of 276 mA/mm and an on-state resistance of 16.6 Ω·mm. Figure 3b shows the I_D_-V_G_ characteristics of the device, on which we can see the threshold voltage (V_TH_) is approximately 1.6 V under the criterion of I_D_ = 0.1 mA/mm. The prepared device has a gate breakdown performance similar to that of a traditional Schottky gate.

In order to monitor the temperature distribution of the device under high power stress, thermal reflectivity imaging was used [28] and the test system setup is shown in Figure 4a. In this work, a 365 nm ultraviolet light-emitting diode light source and a 50× lens were equipped in the measurement system. Before the measurement, the reflectance is calibrated with a thermocouple that needs to be pressed against the sample surface for high accuracy, as shown in Figure 4b. Since the absorption wavelength of GaN material is 362 nm, the 365 nm light source that induces the photovoltaic effect will change the real voltage applied on the ITO gate. To suppress this impact, the frame rate is set to the minimum value of one when the reflectance is sampled.

Figure 5a illustrates the temperature distribution of the ITO-gated HEMTs along the horizontal cutline as depicted on the inset figure, where the V_GS_ = 6 V and V_DS_ = 10/20/30 V. We can unambiguously see that the heat is mainly concentrated in the center of the access area between the gate and drain terminal, which is induced by the electric field peak in the vicinity of the gate field plate edge, consistent with the conclusion reported in [29]. Figure 5b shows the temperature distribution of the ITO-gated HEMT along the vertical cutline from the source terminal’s upper edge to the drain terminal’s bottom edge. It is worthwhile noting that the temperature increase is not significant in the channel area under the gate field plate, thanks to the electric field modulation by the gate field plate. After a 691 s continuous power stress at V_GS_ = 6 V and V_DS_ = 30 V, a failure took place and the gate current saw a sudden jump, as shown in Figure 5c. We can clearly see a hot spot in the inset figure, corresponding to the breakdown point. The cause is probably that, during the p-GaN patterning, the ICP etching introduced enormous defect states on the sidewall of the p-GaN. In addition, the crowding effect of the electric field could accelerate the wear-out process on the sidewall and finally trigger the failure.

To further probe the failure mechanism of the device stressed by the high power shown in Figure 5, gate luminescence was monitored. Figure 6 illustrates that when V_DS_ = 0 V and V_GS_ vary at different voltages, luminescence can be observed from the failed p-GaN gate, thanks to the transparent ITO. It means the p-GaN gate stack works as a light-emitting diode (LED) after the gate failure. Normally, luminescence can be hardly observed because the hole injection from the gate metal to the p-GaN is difficult. However, in our case, the junction between the gate and p-GaN, as well as the p-GaN sidewalls, is quite vulnerable to the high power stress. The most obvious luminescence spot is located on the p-GaN sidewall in the central vicinity of the hottest area in Figure 6. It proves that the gate failure is accelerated by the heat generated by the high power stress. After gate failure, as shown in Figure 7, the Schottky barrier of the gate metal/p-GaN junction is destroyed, and the hole potential barrier is thus lowered, which strongly boosts the hole injection from the gate metal to the p-GaN layer. The abundant injected holes recombine with the electrons from the channel and thus photons are generated and observed.

## 4. Conclusions

In summary, by applying transparent ITO as the gate terminal onto the p-GaN gate HEMTs, the temperature distribution and luminescence of the devices under high power stress were successfully and unambiguously observed by ultraviolet light reflectivity testing. It was clearly concluded that the heat concentrated in the vicinity of the gate field plate in the access area, under the stress of V_GS_ = 6 V and V_DS_ = 10/20/30 V. After a 691 s high power stress, the device failed and the breakdown point was found to be located on the p-GaN. After failure, gate luminescence was found to be distributed on the sidewall of the p-GaN while positively biasing the gate, proving that the sidewall is the weakest spot of the p-GaN gate HEMTs under high power stress. Further optimization of the device processing is suggested to repair the processing damages on the sidewall. These findings above illustrate that the transparent ITO gate is a powerful tool for the reliability characterization of the p-GaN gate HEMTs. This tool can be also be applied in more complicated dynamic reliability tests in the future, considering the ns-level temporal accuracy of the ultraviolet light reflectivity testing system.

## Figures and Tables

**Figure 1 micromachines-14-00940-f001:**
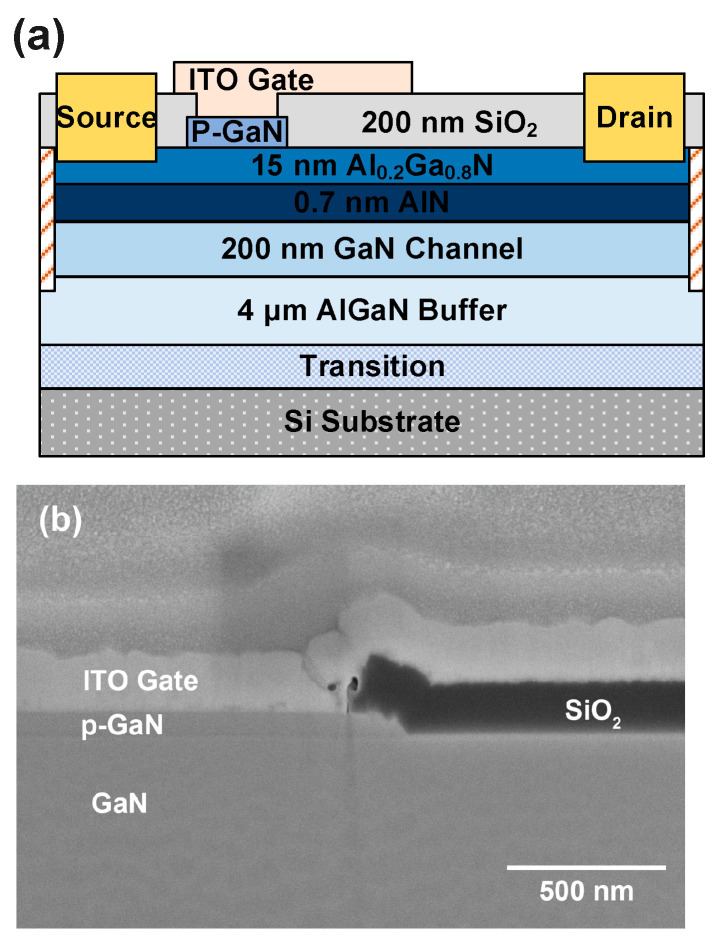
(**a**) Cross-sectional diagram of a p-GaN gate HEMTs with an ITO gate. (**b**) FIB-SEM photograph of the device’s gate stack as well as the gate field plate above the SiO_2_ passivation layer.

**Figure 2 micromachines-14-00940-f002:**
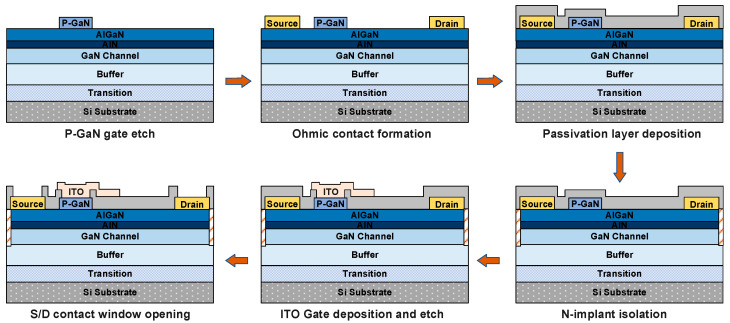
Simplified fabrication process of the ITO-gated p-GaN gate HEMTs.

**Figure 3 micromachines-14-00940-f003:**
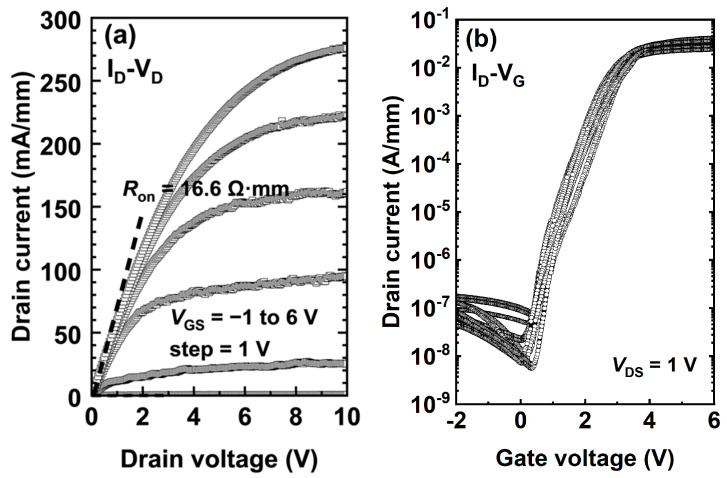
(**a**) Output and (**b**) transfer characteristics of the ITO-gated p-GaN gate HEMTs at room temperature.

**Figure 4 micromachines-14-00940-f004:**
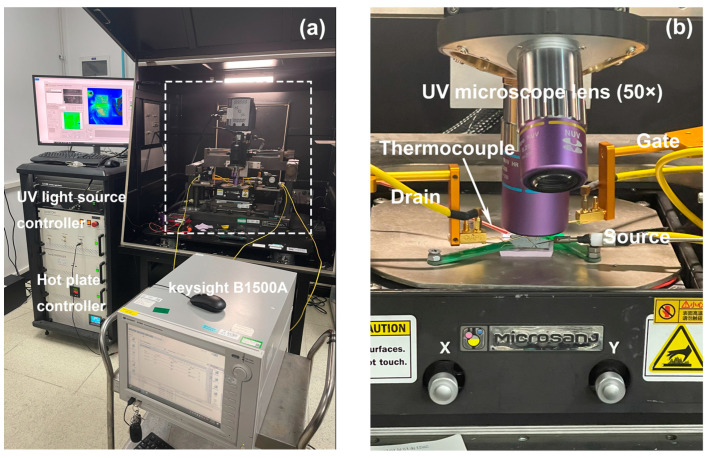
(**a**) The thermal reflectivity imaging system, including electrical measurement, heat measurement, and reflectance calibration section. (**b**) The details of the system dotted box in (**a**).

**Figure 5 micromachines-14-00940-f005:**
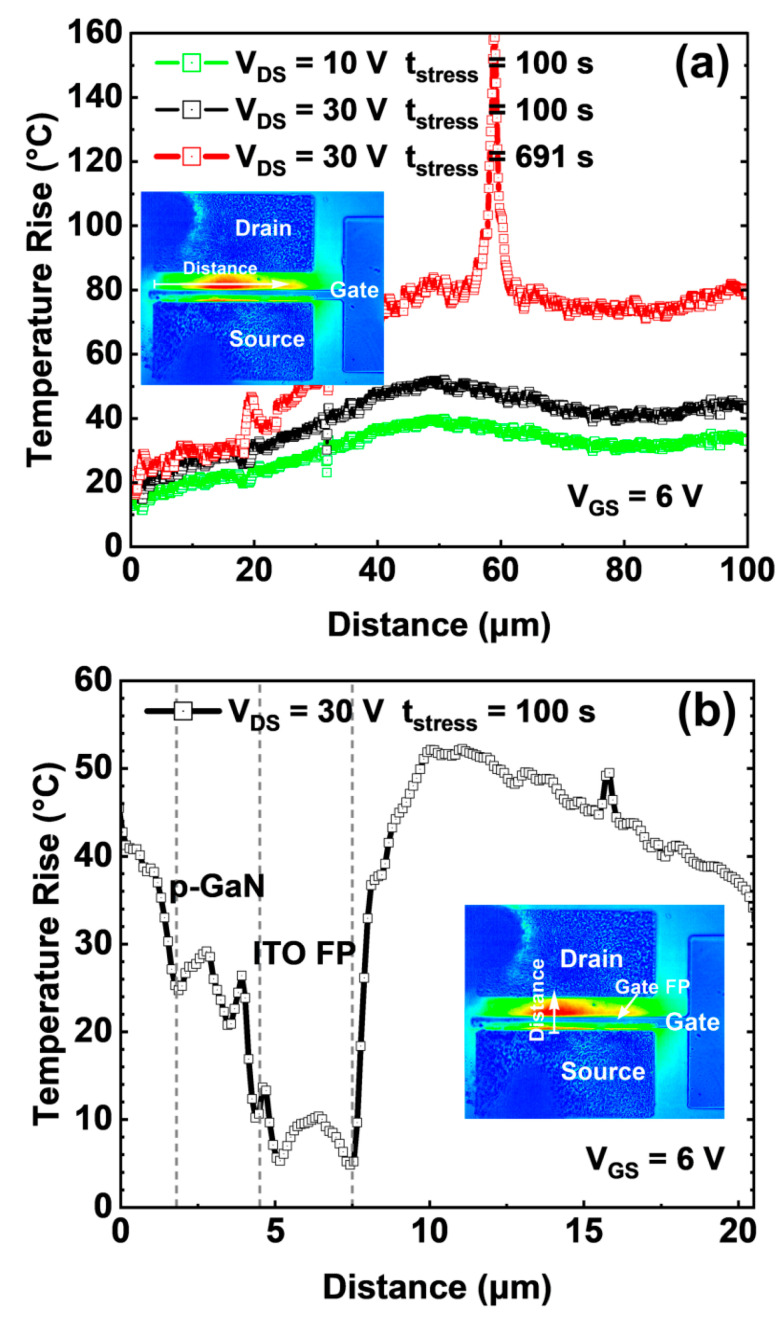
(**a**) The temperature distribution of the ITO-gated HEMTs along the horizontal cutline as depicted on the inset figure, under the stress of V_GS_ = 6 V and V_DS_ = 10/20/30 V. (**b**) The temperature distribution of the ITO-gated HEMTs from the source terminal’s upper edge to the drain terminal’s bottom edge, under the stress of V_GS_ = 6 V and V_DS_ = 30 V by t_stress_ = 100 s. (**c**) The temperature distribution of the ITO-gated HEMTs under the stress of V_GS_ = 6 Vand V_DS_ = 30 V by t_stress_ = 691 s until the failure took place.

**Figure 6 micromachines-14-00940-f006:**
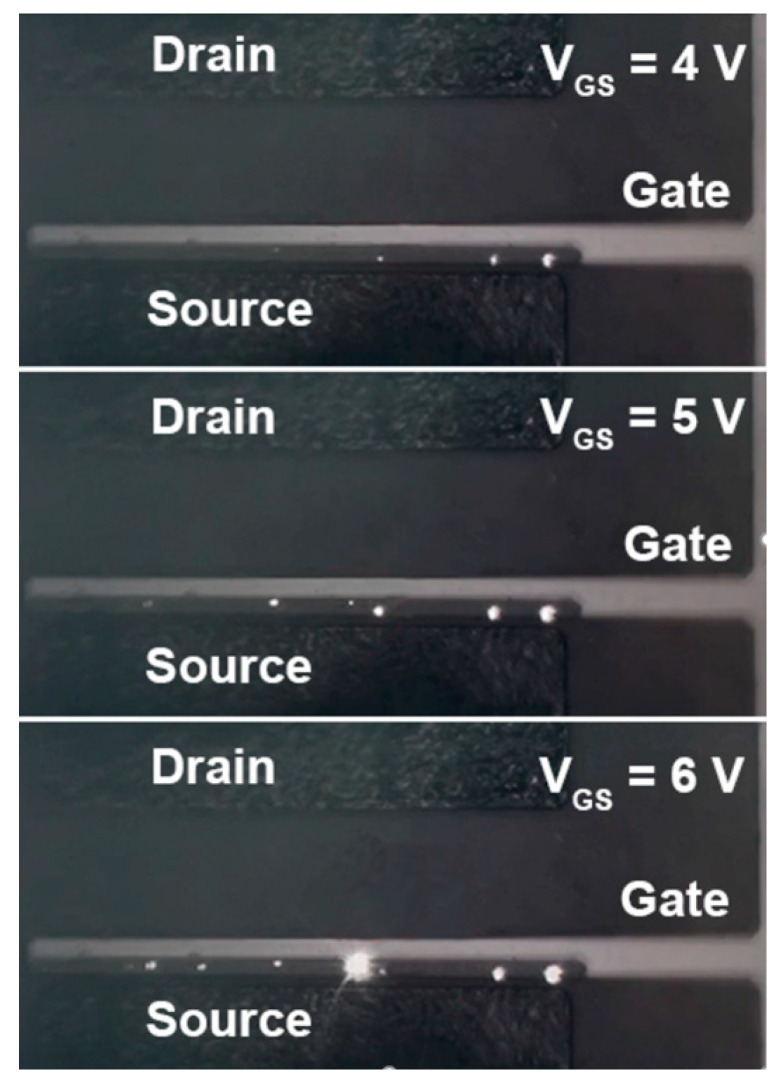
Photos of ITO-gated p-GaN gate HEMTs luminescence under V_DS_ = 0 V and various V_GS_ after gate failure.

**Figure 7 micromachines-14-00940-f007:**
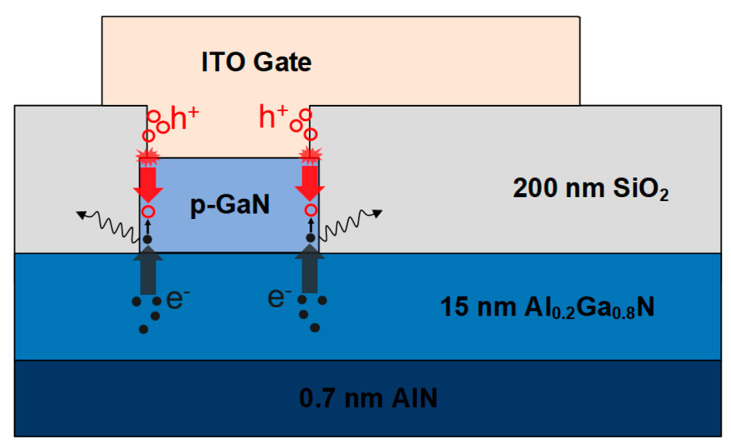
Gate failure and luminescence mechanism of the p-GaN gate HEMTs.

## Data Availability

The data that support the findings of this study are available from the corresponding authors upon reasonable request.

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
