# Peer review of "Investigating the Failure Mechanism of p-GaN Gate HEMTs under High Power Stress with a Transparent ITO Gate"

_micromachines, 2023, doi:10.3390/mi14050940_

Round 1

Reviewer 1 Report

This paper presents the development of a HEMT GaN-AlGaN transistor with a transparent gate electrode. This allows us to more accurately investigate the temperature distribution in the transistor core. The article looks quite original and well written. Regarding the article, I had only one question - why did the authors use a silicon substrate for growth, and not silicon carbide? First, SiC has a smaller lattice mismatch with GaN than Si. This leads to a better structural perfection of the grown GaN/AlGaN layers. Secondly, silicon carbide has a significantly higher thermal conductivity than silicon. The use of a SiC substrate would reduce the effect of heat on device performance. After the authors clarify this point, the article can be accepted for publication.

I think that English Language is OK

Author Response

Thank you for your careful reviewing and helpful comments, which will greatly improve the quality of our manuscript. Considering your comment on why you don't use a better SiC substrate, we mainly consider the cost issue. In mass production, cost is a key consideration, and the price of GaN-on-SiC is much higher than GaN-on-Si. Therefore, GaN-on-Si is preferred in consumer electronics such as fast charging.

Reviewer 2 Report

This paper entitled with ‘Investigating the failure mechanism of p-GaN gate HEMTs under high power stress with a transparent ITO gate’ has demonstrated a HEMT devices with ITO as gate materials, which is useful for defect/failure detection since ITO is transparent compared with conventional metal. Even though ITO as gate material is not a novel concept, the authors have found a point that it helps find to the failure points, which is quite interesting for the community. Overall, the paper is well-written, well-referenced, and well-organized. I recommend this manuscript to be accepted for publication after addressing the following questions.

1.       Regarding the HEMT structure, is there any AlN interlayer between AlGaN barrier and GaN layer?

2.       Any annealing for ITO gate metal?

3.       In L75 said that ‘ITO was deposited via evaporation at 350 C Afterwards’. I wonder what the 350 C is temperature? Is it the substrate temperature or the target temperature?

4.       There should be more discussions on Figure 6, which is the main advantage this paper wants to present. Are these two photos from the same device? Where are the exact position of these points of failure? Are there any correlations between these points to the temperature distribution discussed in the previous session?

5.       Is there a specific reason to choose 691 s as high-power stress duration?

Author Response

Response:

First, thank you for your careful reading and meticulous comment, and thank you very much for pointing out the negligence of our work. Below I will explain each of your valuable comments:

  1. There is a 0.7 nm AlN insertion layer in the HEMT structure, which we have corrected in the manuscript and thank you for pointing out the problem.
  2. In this study, the ITO gate did not undergo a post-contact annealing process like the traditional Ni/Au Schottky gate.
  3. In L75, 350°C refers to the temperature of the substrate, which has been modified in the article to prevent misunderstandings. Regarding the formation of ITO film, we use an optical coating machine, in which the entire cavity is in a high temperature state during the process, and the substrate is maintained at 350°C.
  4. The device measured in Figure 6 is exactly the device in Figure 5 after high power stress. The failures take place on the sidewall of the p-GaN and the junction between the gate metal and p-GaN. The failure spot is in vicinity of the central hottest area, proving that the gate failure is directly accelerated by the heat generated by the high power stress. To clearly analyze the failure mechanism, we added figure 7 to the revised manuscript. The following explanation has been supplemented to the revised manuscript.

(However, in our case, the junction between the gate and p-GaN as well as the p-GaN sidewalls are quite vulnerable to the high power stress. The most obvious luminescence spot is locating on the p-GaN sidewall in vicinity of the central hottest area in Figure 6. It proves that the gate failure is accelerated by the heat generated by the high power stress. After gate failure as shown in Figure 7, the Schottky barrier of the gate metal/ p-GaN junction is destroyed, the hole potential barrier is thus lowered, which strongly boosts hole injection from the gate metal to the p-GaN layer. The abundant injected holes recombine with the electrons from the channel and thus photons are generated and observed.)

  1. 691s is the time when the device gate fails, and the device fails under the stress conditions of VGS = 6 V and VDS = 30 V, and the heat distribution at this time is automatically recorded by the heat reflection system.

Reviewer 3 Report

Han et al. in the manuscript "Investigating the failure mechanism of p-GaN gate HEMTs under high power stress with a transparent ITO gate" presented important results. The research approach is innovative and have great potential.

Recommend minor revision with the following inclusion!

Breakdown voltage

Transport properties that include the charge mobility

added discussion on reliability with references.

Author Response

Response:

Thank you for your helpful comments on the refinement and enhancement of our manuscript, which we have supplemented with breakdown voltage data,transport properties and references on reliability.